# Moderate and Stable Pain Reductions as a Result of Interdisciplinary Pain Rehabilitation—A Cohort Study from the Swedish Quality Registry for Pain Rehabilitation (SQRP)

**DOI:** 10.3390/jcm8060905

**Published:** 2019-06-24

**Authors:** Åsa Ringqvist, Elena Dragioti, Mathilda Björk, Britt Larsson, Björn Gerdle

**Affiliations:** 1Department of Neurosurgery and Pain Rehabilitation, Skåne University Hospital, SE-221 85 Lund, Sweden; asa.ringqvist@skane.se; 2Pain and Rehabilitation Centre, and Department of Medical and Health Sciences, Linköping University, SE-581 85 Linköping, Sweden; elena.dragioti@liu.se (E.D.); britt.larsson@liu.se (B.L.); 3Department of Social and Welfare Studies, Linköping University, SE-602 21 Norrköping, Sweden; mathilda.bjork@liu.se

**Keywords:** chronic pain, musculoskeletal pain, patient care team, rehabilitation, treatment outcome

## Abstract

Few studies have investigated the real-life outcomes of interdisciplinary multimodal pain rehabilitation programs (IMMRP) for chronic pain. This study has four aims: investigate effect sizes (ES); analyse correlation patterns of outcome changes; define a multivariate outcome measure; and investigate whether the clinical self-reported presentation pre-IMMRP predicts the multivariate outcome. To this end, this study analysed chronic pain patients in specialist care included in the Swedish Quality Registry for Pain Rehabilitation for 22 outcomes (pain, psychological distress, participation, and health) on three occasions: pre-IMMRP, post-IMMRP, and 12-month follow-up. Moderate stable ES were demonstrated for pain intensity, interference in daily life, vitality, and health; most other outcomes showed small ES. Using a Multivariate Improvement Score (MIS), we identified three clusters. Cluster 1 had marked positive MIS and was associated with the overall worst situation pre-IMMRP. However, the pre-IMMRP situation could only predict 8% of the variation in MIS. Specialist care IMPRPs showed moderate ES for pain, interference, vitality, and health. Outcomes were best for patients with the worst clinical presentation pre-IMMRP. It was not possible to predict who would clinically benefit most from IMMRP.

## 1. Introduction

Pain is an unpleasant experience with complex interactions between sensorimotoric, affective, and cognitive brain networks. As such, pain, especially chronic pain, is influenced by and interacts with physical, psychological, social, and contextual factors [1,2,3]. One-fifth of the European population has moderate to severe chronic pain conditions [4]. These conditions are associated with psychological distress, low health, sick leave, and high socioeconomic costs [5]. Therefore, a biopsychosocial (BPS) framework should be considered in clinical practice [6,7,8]. 

Unlike single/unimodal interventions, interdisciplinary multimodal pain rehabilitation programs (IMMRPs) for chronic pain—an interdisciplinary treatment according to the International Association for the Study of Pain (IASP)—distinguish themselves as well-coordinated complex interventions. Typically, IMMRPs are based on cognitive behavioural therapy (CBT) models (including Acceptance Commitment Therapy, ACT) and are administered over several weeks to months [9,10,11,12]. The Swedish programs generally include group activities such as pain education, supervised physical activity, training in simulated environments, and CBT coordinated by an interdisciplinary team (e.g., physician, occupational therapist, physiotherapist, psychologist, and social worker) based on a BPS framework [9,10,11,12]. The components of IMMRP are most often chosen based on the available evidence for unimodal interventions for chronic pain, for example, with respect to education, exercise, psychological interventions, and interventions for return to work. The core goals of rehabilitation programs in general [13] and especially for patients with chronic pain [14] are broad and multifactorial in combination with the individualised goals of the patient. These include increased ability to participate in valued activities such as work. Hence, IMMRP is a complex intervention [13,15] and, unlike pharmacological intervention, focusses on the whole person rather than just biochemical processes, implying complex patient conditions matched with complex IMMRPs [16,17]. The components of IMMRP can be active independently or interdependently [15], resulting in a combination of effects explained by known and unknown mechanisms. The effects are assumed to be greater than the sum of its components [18]. 

Systematic reviews (SRs) have generally reported higher efficacy both on a general level and for specific outcomes of IMMRP compared with single-treatment or treatment-as-usual programs [10,12,19,20,21,22,23]. SRs and Randomised Controlled Trials (RCTs) may be associated with risk for bias resulting from, for example, an unrepresentative selection of patients and researcher allegiance [24,25,26]. Thus, it is necessary to investigate whether the evidence obtained from SRs and RCTs can be replicated within a consecutive non-selected flow of patients in practice settings using prospective observational cohort study designs such as practice-based evidence (PBE). PBE has also been applied in the field of rehabilitation research [27]. The importance of such an approach is also emphasised in the real-effectiveness medicine framework [28]. IMMRPs are time consuming and expensive, even when most of the activities are group-based. From an ethical, individual, and socioeconomic perspective, it is indeed remarkable to note the lack of studies investigating effect sizes (ES) in patient populations in real-life practice settings. A recent study from two Swedish university clinics reported effect sizes of 0.51–0.61 (i.e., moderate ES) for two pain intensity variables at 12-month follow-up [29]. These effect sizes should be confirmed in larger studies based not only on patients at university hospitals, but also on specialist units in general. It would be motivating for patients to endure increases in pain, which is often observed in clinical practise during the start-up period of rehabilitation characterised by an increase in activity levels, if it were known that the long-term effects include the reduction of pain levels.

Complex interventions such as IMMRP should have several outcomes measured at multiple levels and strategies for handling multiple outcomes [17,30]. IMMRPs are evaluated using many outcomes. For example, one SR including 46 RCTs reported nine outcomes per RCT (median) [10]. However, outcomes are not usually divided into primary and secondary outcomes [10]. In addition, although it is most likely that changes in several of the selected outcomes are correlated, most SRs of IMMRPs evaluate the outcomes as independent from each other. Patterns of potential correlations (i.e., multivariate correlation patterns) are mainly unknown/uninvestigated, even though they could give valuable information regarding how to optimise IMMRPs. Hence, there is a need to develop clinically applicable ways to evaluate the multiple outcomes of MMPRs both for individual patients and within research studies. 

The above knowledge gaps motivated this PBE study of chronic pain patients based on patient reported outcome measures (PROMs) from the Swedish Quality Registry for Pain Rehabilitation (SQRP) [31]. This registry offers an opportunity to investigate clinical outcomes and patterns of change, since all the relevant specialist care units throughout Sweden deliver data to SQRP. Hence, this PBE study has the general aim of investigating the effects of IMMRP in specialist care in Sweden considering the multivariate complexity of outcomes. We hypothesised that IMMRP in special care is associated with small-to-medium ES, that changes in outcomes generally are intercorrelated, and that the baseline situation (pre-IMMRP) can predict the multivariate outcomes. More specifically, we defined the following four aims:To investigate the outcome effect sizes of IMMRP immediately post-IMMRP and at 12-month follow-up.To analyse the multivariate correlation patterns of changes in outcomes of IMMRP: pre-IMMRP versus post IMMRP and pre-IMMRP versus 12-month follow-up.To define a multivariate outcome measure of IMMRP.To investigate if the clinical self-reported presentation pre-IMMRP can predict the multivariate outcome measure.

## 2. Materials and Methods

### 2.1. The Swedish Quality Registry for Pain Rehabilitation (SQRP) 

The SQRP, recognised by the Swedish Association of Local Authorities and Regions, receives data from all specialist care units in Sweden [31]. The SQRP is based on PROM questionnaires that capture biopsychosocial data such as the patient’s background, pain distribution and intensity, pain-related cognitions, and psychological distress symptoms (e.g., depression and anxiety), as well as activity/participation aspects and health-related quality of life variables. Patients complete the PROM questionnaires on up to three occasions: (1) during assessment at the first visit to the unit (pre-IMMRP); (2) immediately after the IMMRP (post-IMMRP); and (3) at the 12-month follow-up (FU) after IMMRP discharge (12-month FU).

### 2.2. Subjects

This study included SQRP data from women and men ≥18 years old with complex chronic (≥3 months) non-malignant pain who were referred to specialist pain and rehabilitation units (i.e., specialist care centres) between 2008–2016. These patients can be characterised as complex, as their health profiles included psychiatric comorbidities such as depression and anxiety, low levels of acceptance, high levels of kinesiophobia, decreased working life and participation in social activities, and/or did not respond to routine pharmacological/physiotherapeutic treatments delivered in a monodisciplinary fashion. Strict inclusion and exclusion criteria for inclusion in the registry is not available, since this is a registry study of patients with complex chronic pain conditions referred from mainly the primary care to specialist care in Sweden. A minority of patients were referred from other specialist clinics e.g., orthopedic and rheumatology clinics. The following general inclusion criteria for IMMRP were used: (i) disabling chronic pain (on sick leave or experiencing major interference in daily life due to chronic pain); (ii) age 18 years and above; (iii) no further medical investigations needed; and (iv) written consent to participate and attend IMMRP. General exclusion criteria for IMMRP included severe psychiatric morbidity, abuse of alcohol and/or drugs, diseases that did not allow physical exercise, and specific pain conditions with other treatment options available (i.e., red flags).

The proportions of patients within primary health care with chronic pain conditions are not exactly known, but 10–20% are estimates [32,33]. Furthermore, the proportion of chronic pain patients within primary health care that are referred to specialist clinics is not known.

The study was conducted in accordance with the Helsinki Declaration and Good Clinical Practice and approved by the Ethical Review Board in Linköping (Dnr: 2015/108-31). All the participants received written information about the study and gave their written consent.

### 2.3. Variables

Background variables that were collected pre-IMMRP and symptom-related self-reported variables that were collected at all three times (pre, post, and 12-month FU) were used in the analyses. The variables and instruments used are mandatory for the units registering their data with the SQRP.

#### Background Variables

The following background variables were collected: age (years), gender (man or woman), education level, and country of birth. Education level was dichotomised into university and the other alternatives (i.e., upper secondary school, elementary school, or other); this variable was labelled as University. Country of birth was dichotomised as from Europe and outside Europe and labelled as Outside-Europe. In addition, self-reported pain duration (days), persistent pain duration (days), and number of days off work (Days no work) were obtained.

Pain distribution was registered using 36 predefined anatomical areas (18 on the front and 18 on the back of the body) and the patients registered the areas with pain: (1) head/face, (2) neck, (3) shoulder, (4) upper arm, (5) elbow, (6) forearm, (7) hand, (8) anterior aspect of chest, (9) lateral aspect of chest, (10) belly, (11) sexual organs, (12) upper back, (13) low back, (14) hip/gluteal area, (15) thigh, (16) knee, (17) shank, and (18) foot. The number of areas with pain (range: 1–36) were summed, and the obtained variable was denoted as the Pain Region Index (PRI).

### 2.4. Repeated Self-Reported Measures

For reports of the psychometric aspects of the self-reported measures, the reader is referred to other studies summarising these [7,34,35,36].

#### 2.4.1. Pain Aspects

Pain intensity average during the previous seven days was registered using a 0–10 (0 = no pain and 10 = worst possible pain) numeric rating scale (NRS)—NRS-7days.

#### 2.4.2. The Multidimensional Pain Inventory (MPI)

MPI is a 61-item self-report questionnaire that measures the psychosocial, cognitive, and behavioural effects of chronic pain [37,38]. Part 1 consists of five scales: Pain severity—measuring several aspects of the pain experience (MPI-Pain-severity); Interference—pain-related interference in everyday life (MPI-Pain-interfer); Perceived Life Control (MPI-LifeCon); the level of affective distress (MPI-Distress); and Social Support—perceived support from a spouse or significant others (MPI-SocSupp). Part 2 assesses the perception of responses to displays of pain and suffering from significant others and consists of three scales: Punishing Responses (MPI-Punish); Solicitous Responses (MPI-Solict); and Distracting Responses (MPI-Distract). Part 3 measures to what extent the patients participate in various activities using four scales. These scales can be combined into a composite scale—the General Activity Index (MPI-GAI)—which was used in the present study [39].

#### 2.4.3. Psychological Distress Variables

Symptoms of anxiety and depression were registered using the Hospital Anxiety and Depression Scale (HADS) [40,41]. This instrument comprises seven items in each of two subscales: depression (HADS-D) and anxiety (HADS-A) symptoms. Both subscale scores have a range of 0 to 21. A score of 7 or less in each subscale indicates a non-case, a score of 8–10 indicates a possible case, and a score of 11 or more indicates an almost definite case [40]. 

#### 2.4.4. The Short Form Health Survey (SF36)

The Short Form Health Survey (SF36) attempts to represent multidimensional health concepts and measurements of the full range of health states, including levels of well-being and personal evaluations of health [42]. Scores are standardised into eight dimensions with a scale from 0 to 100 where higher scores indicate a better perception of health [42]: (1) physical functioning (sf36-pf), physical activity level including activities of daily living; (2) role limitations due to physical functioning (sf36-rp), to what extent physical health limits the performance of work and other regular activities; (3) bodily pain (sf36-bp), pain and related disability; (4) general health (sf36-gh), evaluation of health situation; (5) vitality (sf36-vt), how rested and energetic; (6) social functioning (sf36-sf), disturbances of social life due to physical or mental illness; (7) role limitations due to emotional problems (sf36-re), difficulties in performing work or other regular activities due to emotional problems; and (8) mental health (sf36-mh), anxiety and depressive symptoms. Based on the eight scales, a physical summary component and a mental (psychological) summary component can be calculated, but these two summary component variables were not used in the present study.

#### 2.4.5. The European Quality of Life Instrument (EQ-5D)

The European Quality of Life Instrument (EQ-5D) captures a patient’s perceived state of health [43,44,45]. The first part of the instrument defines five dimensions: mobility, self-care, usual activities, pain/discomfort, and anxiety/depression. Each of these were measured at three levels. An EQ-5D-index is derived by applying a formula that essentially attaches values (weights) to each of the levels in each dimension. The collection of index values (weights) for all the possible EQ-5D states is called a value set. Most EQ-5D value sets have been obtained from a standardised valuation exercise where a representative sample of the general population in a country/region is asked to place a value on EQ-5D health states. The EQ5D also measures the self-estimation of today’s health according to a 100-point scale, which is a thermometer-like scale (EQ-VAS) with defined end points (high values indicate good health and low values indicate bad health). 

#### 2.4.6. Estimations of Changes in Pain and in Life Situation 

The patients post-IMMRP and at the 12-month FU estimated the degree of positive change in pain (Change-pain) and in their ability to handle life situations in general (Change-life situation). The Change-pain item was rated on a five-point Likert scale from markedly increased pain (0) to markedly decreased pain (4). The Change-life situation item was rated on a five-point Likert scale from markedly worsened (0) to markedly improved (4). 

### 2.5. Statistics

All the statistics were performed using the statistical package IBM SPSS Statistics (version 24.0) and SIMCA-P+ (version 15.0; Umetrics Inc., Umeå, Sweden). A probability of <0.001 (two-tailed) was accepted as the criteria for significance due to the large number of subjects. 

The text and tables generally report the mean value ± one standard deviation (±1 SD) together with a median and range of continuous variables. Percentages (%) are reported for categorical variables. The detailed analyses also report 95% confidence intervals (95% CI). SQRP uses predetermined rules when handling single missing items of a scale or a subscale; details about this have been reported elsewhere [29]. To compare groups, we used Student´s *t*-test for unpaired observations, analysis of variance (ANOVA with post hoc test if significant difference), and Chi square test. Effect sizes (ES; Cohen’s d) for within-group analysis were computed using a calculator when appropriate (https://webpower.psychstat.org/models/means01/effectsize.php). Hedges’ g, which provides a measure of effect size weighted according to the relative size of each sample, was used for between ES using a calculator (https://www.socscistatistics.com/effectsize/default3.aspx). The absolute effect size was considered very large for values ≥ 1.3, large for values between 0.80–1.29, moderate for values between 0.50–0.79, small for values between 0.20–0.49, and insignificant for values < 0.20 [46].

Common methods such as logistic regression (LR) and multiple linear regression (MLR) can quantify the level of relations of individual factors but disregard interrelationships among different factors and thereby ignore system-wide aspects [47]. Moreover, such methods assume variable independence when interpreting results [48], and there are several risks when considering one variable at a time [49]. To account for our aims, the problems related to handling missing data (see below), and the risks associated with multicollinearity problems (see above), we used advanced multivariate analyses (MVDA). 

Hence, using SIMCA-P+, we applied advanced Principal Component Analysis (PCA) for the multivariate correlation analyses of all investigated variables and Orthogonal Partial Least Square Regressions (OPLS) for the multivariate regressions. These techniques do not require normal distributions of the included variables [50]. Note that the PCA of SIMCA-P+ differs considerably from the simpler version implemented (e.g., the version used in SPSS). 

PCA extracts and displays systematic variation in the data matrix. All the variables were log transformed before the statistical analyses if data were skewed. Using PCA, we analysed the multivariate correlation pattern for the changes in the 22 outcome variables for all the subjects. Note that changes in outcomes are calculated so that a positive value indicates an improvement. A cross-validation technique was used to identify nontrivial components (p). Variables loading on the same component p were correlated, and variables with high loadings but with opposing signs were negatively correlated. Variables with high absolute loadings were considered significant. The obtained components are per definition not correlated and are arranged in decreasing order with respect to explained variation. The loading plot reports the multivariate relationships between variables. A corresponding plot reporting the relationships between subjects (i.e., *t*-scores) can also be used (score plot), and each subject receives a score (*t*) for each of the significant components. The *t*-score was used to calculate a Multivariate Improvement Score (MIS). R^2^ describes the goodness of fit—the fraction of sum of squares of all the variables explained by a principal component [51]. Q^2^ describes the goodness of prediction—the fraction of the total variation of the variables that can be predicted using principal component cross-validation methods [51]. Outliers were identified using two methods: score plots in combination with Hotelling’s T^2^ and distance to model in the X-space. No extreme outliers were detected. 

OPLS was used for the longitudinal multivariate regression analyses of the *t*-scores of the PCA mentioned above using pre-IMMRP data (i.e., baseline data) [51]. The variable influence on projection (VIP) indicates the relevance of each X-variable pooled over all dimensions and Y-variables—the group of variables that best explain Y. VIP ≥ 1.0 was considered significant if VIP had a 95% jack-knife uncertainty confidence interval non-equal to zero. p(corr) was used to note the direction of the relationship (positive or negative). This is the loading of each variable scaled as a correlation coefficient, and thus standardising the range from −1 to +1. [50] p(corr) is stable during iterative variable selection and comparable between models [50]. Thus, a variable/regressor was considered significant when VIP > 1.0. For each regression, we report R^2^, Q^2^, and the *p*-value of a cross-validated analysis of variance (CV-ANOVA). SIMCA-P+ uses the Non-linear Iterative Partial Least Squares (NIPALS) algorithm to handle missing data: maximum 50% missing data for variables/scales and maximum 50% missing data for subjects. 

To identify clusters based on the *t*-scores of the PCA mentioned above, we performed hierarchical clustering analysis (HCA). Based on the identified clusters (subgroups) defined by HCA, we performed partial least squares discriminant analysis (PLS-DA). In addition, we applied a bottom–up HCA to the principal component score vectors using the default Ward linkage criterion to identify relevant subgroups of patients. HCA can find subtle clusters in the multivariate space. In the resulting dendrogram, clusters were identified and, based on these groups, we performed PLS-DA using group belonging as the Y-variable and the psychometric data as predictors (X-variables). The PLS-DA model was computed to identify associations between the X-variables and the subgroups. Based on the HCA defined clusters, traditional inferential statistics (ANOVA including post hoc tests when appropriate) were computed using SPSS. 

## 3. Results

### 3.1. Background Data

There were 14,666 chronic pain patients registered in the SQRP that fulfilled the inclusion criteria: chronic pain; >18 years of age; and completed the SQRP questionnaire before and on at least one of the two time points after the IMMRP. More than half (60%) of the patients answering the questionnaires pre-IMMRP and post-IMMRP also answered the questionnaires at 12-m FU. Most of the patients (76.3%) were women, 25.2% had studied at university, and 10.4% were born outside of Europe. More men were born outside Europe (men: 13.4% versus women: 9.5%; Chi^2^ = 43.437, *p* < 0.001), and fewer men had university education (men: 18.0% versus women: 27.4%; Chi^2^ = 123.672; *p* < 0.001). Continuous background variables are shown in Table 1. Women were slightly younger than men (42.9 ± 10.7 versus 44.5 ± 10.7; *p* < 0.001) and reported more spreading of pain according to PRI (15.4 ± 8.8 versus 10.8 ± 7.0; *p* < 0.001). The other variables in Table 2 were not affected by gender.

### 3.2. Pairwise Comparisons of Repeated Measures

The results for pre-IMMRP and post-IMMRP are shown in Table 2. Significant improvements were generally found except for two of the three scales of the second part of the MPI. In addition, the comparisons between pre-IMMRP and the 12-month FU generally revealed significant improvements except for one of the scales on the second part of the MPI (Table 3). Some outcomes were associated with moderate effect sizes. For the pre-IMMRP versus post-IMMRP comparisons, three variables had moderate effects sizes: MPI-pain-severity, sf36-bp, and sf36-vt (Table 2). At the 12-month FU, MPI-pain-severity and sf36-bp were associated with moderate effect sizes; this was also the case for MPI-pain-interference and EQ5d-index (Table 3). However, generally small effect sizes were found for the significant improvements (Table 2 and Table 3). The variables of the second part of the MPI had insignificant effect sizes both post-IMMRP and 12-month FU. 

### 3.3. Patients Not Participating in the 12-Month FU

There were only small differences between those reporting PROM data at the 12-month FU and those not reporting their situation pre-IMMRP (Appendix A). Although those not reporting had a somewhat worse situation for most of the PROM variables, the differences were of no clinical importance. 

### 3.4. Estimations of Changes in Pain and in Life Situation 

At both post-IMMRP and 12-month FU, most patients reported that their pain situation had improved as well as their ability to handle their life situation (Table 4). 

### 3.5. Multivariate Correlation Pattern of Changes in Outcomes

PCAs of the changes (i.e., the difference) were performed for pre-IMMRP versus post-IMMRP and pre-IMMRP versus 12-month FU. Significant models were achieved for both analyses (Table 5). Similar patterns were obtained for the first significant component of the two PCAs (Table 5). The first component of both analyses, reflecting the most important variations, showed that changes in HAD-D, MPI-pain-severity, MPI-pain interference, MPI-control, MPI-distress, sf-36-bp, sf-36-vt, sf-36-sf, and sf36-mh were most important and intercorrelated significantly. Hence, it was obvious that the changes in outcome variables are intercorrelated. That is, rather than representing 22 independent variables, the multivariate analyses show that most changes in these variables are highly intercorrelated. 

At 12-month FU, the PCA also identified two additional components (Table 5). The second component mainly reflected the intercorrelation pattern between the social support scale of the MPI and the scales of part 2 of the MPI. A third significant component only explaining 6% of the variation in the dataset was also obtained in the analysis of changes at the 12-month follow-up versus baseline (Table 5). 

The loading plot (i.e., the intercorrelations between variables of the two most important components for the changes pre IMMRP versus 12-month FU) is shown in Figure 1 (Figure 1a is a graphic presentation of the first two components reported in Table 5). Figure 1b shows the corresponding score plot (i.e., the relationships between subjects/patients). Each patient can be described with a score (*t*-score) for each significant component. Patients with high positive *t*-scores on the first component show prominent changes in the important variables constituting the first component, whereas patients near zero do not benefit, and patients with negative *t*-scores (located to the left in the score plot) deteriorate in the multivariate context. Hence, the *t*-score of the first component of both analyses can be considered as a Multivariate Improvement Score, in the following denoted MIS-post-IMMRP and MIS-12-month FU. 

### 3.6. Identification of Subgroups Based on the Multivariate Improvement Scores (MIS) 

An HCA based on MIS-post-IMMRP was performed. Three subgroups/clusters were identified. Based on the HCA, a PLS-DA model with two predictive components was obtained with group belonging as the Y-variable (R^2^ = 0.35; Q^2^ = 0.35; CV-ANOVA *p* < 0.001; *n* = 14,666). Using a similar approach, we performed an HCA based on MIS-12-month FU. This analysis identified three subgroups/clusters. Based on the HCA, a PLS-DA model with two predictive components was obtained with group belonging as the Y-variable (R^2^ = 0.37; Q^2^ = 0.37; CV-ANOVA *p* < 0.001; *n* = 8851).

The MIS (i.e., *t*-score) showed clear positive values (i.e., improvements) for cluster 1 and negative scores (i.e., deterioration) for cluster 3 (Table 6 and Table 7). Cluster 2 was an intermediary cluster with overall slightly positive improvements. Prominent effect sizes in the pairwise comparisons were observed post-IMMRP: cluster 1 versus cluster 2 = 3.33; cluster 1 versus cluster 3 = 5.36; and cluster 2 versus cluster 3 = 2.77; 12-month FU: cluster 1 versus cluster 2 = 2.92; cluster 1 versus cluster 3 = 4.99; and cluster 2 versus cluster 3 = 2.34. Thus, distinct differences in improvement levels were detected between the three clusters.

To facilitate the understanding of the identified clusters, the clusters were compared for the variables in each PCA (Table 6 and Table 7). The three clusters differed significantly for all changes according to the ANOVAs performed. The post hoc tests showed that 20 of the 22 changes post-IMMRP differed significantly between all three clusters. The corresponding figure at 12-month FU was 21 of 22 changes.

The estimations of changes in pain (Change-pain) and in management of life (Change-life situation) were not included in the PCAs and thus not included in the calculations of MIS. However, these estimations showed a similar pattern: the most prominent positive changes were in cluster 1, and the least positive changes were in cluster 3 (Table 6 and Table 7). 

In the next step, the identified three clusters of both analyses were compared for their pre-IMMRP values (Table 8 and Table 9). For the clusters obtained post-IMMRP (Table 8), small irrelevant cluster differences existed for age. The proportion with university education was significantly highest in cluster 1 and lowest in cluster 3, although the differences were small. Generally, cluster 1 was associated with the worst situation for the most variables followed longitudinally except for social support, two of the scales of Section 2 of the MPI, and sf36-pf. In contrast, cluster 3 had the best situation, and cluster 2 was intermediate (Table 8). A very similar pattern was found when using the clusters obtained from the 12-month FU (Table 9). 

### 3.7. Longitudinal Regression of MIS Using Baseline Data

The outcome data at baseline (pre-IMMRP) together with the background variables were used to regress MIS-post-IMMRP and MIS-12-month FU (Table 10). For both MIS, psychological distress variables were the most important regressors, but life impact variables, pain aspects, and health and vitality aspects contributed significantly. The directions of the correlations revealed that a more severe clinical situation (e.g., psychological distress, lack of control, low vitality and health, pain interference, and high pain intensity) were associated with high MIS (i.e., multivariate improvements). Although the obtained regressions were highly significant according to the CV-ANOVA, the explained variations in MIS were less than 10% (R^2^ = 0.08 in both analyses). Hence, most of the variations in the two MIS were not possible to predict.

Similar analyses for each of the clusters (Appendix A) revealed that regressions were highly significant, but only explained a minority of the variations in MIS. Although the relative importance of the variables pre-IMMRP differed somewhat between the three clusters, no marked differences existed; that is, psychological distress aspects were the most important post-IMMRP (Appendix A). For the 12-month FU, somewhat more pronounced differences existed between the clusters: in cluster 2, the pain intensity aspects were the most important for MIS, and in cluster 1 and cluster 3, psychological distress variables together with pain interference were the most important for MIS. 

## 4. Discussions

The major findings of the present large PROM study from SQRP are listed below:Moderate long-term ES were found for pain intensity (MPI Pain severity and SF-36 bodily pain), interference in daily life (MPI Interference), and state of health (EQ-5D-index); most other variables showed small ES. Vitality also displayed moderate effect sizes immediately after IMMRP but fell slightly under cut-off for moderate change at 12-month follow-up. The majority of the 22 investigated outcomes were significantly improved.Significant intercorrelations between changes in pain intensity, interference, control, psychological distress, and mental health were confirmed. The changes in 22 outcomes reflected one (pre-IMMRP versus post-IMMRP) or three (pre-IMMRP versus 12-month follow-up) latent components (groups of variables).The outcomes were best for patients with the worst self-reported clinical presentation pre-IMMRP. Based on a defined multivariate improvement score (MIS), three clusters were identified. Cluster 1—overall, the worst situation pre-IMMRP—showed positive multivariate improvements in outcomes. Cluster 3—deteriorated—showed negative scores. Cluster 2, the intermediate cluster, was associated with overall slightly positive multivariate improvements.Certain variables (especially psychological distress and life impact variables, pain, and health and vitality aspects) pre-IMMRP were associated with improvements according to MIS both post-IMMRP and at 12-month FU. However statistically significant, the pre-IMMRP situation could only explain a small part of the variation in MIS (8%); therefore, for clinical use, it was not possible to predict those who would benefit most from IMMRP.

The outcome variables mandatory in SQRP and presented in the present study are in good agreement with the BSP model of chronic pain and the outcome domains presented by the Initiative on Methods, Measurement, and Pain Assessment in Clinical Trials (IMMPACT) [7,52] and the Validation and Application of a patient-relevant core set of outcome domains to assess multimodal PAIN therapy (VAPAIN) [14] initiatives.

The present study was not primarily designed to evaluate the efficacy of IMMRP, which requires RCTs and SRs/meta-analyses. However, our results for the repeated measurements (Table 2 and Table 3) of chronic pain patients in real-life practice settings are in agreement with the positive evidence for IMMRPs reported in SRs [10,11,12] and in other studies [22,23,53]. As such, the small to moderate ES are noteworthy as these patients, who receive pain rehabilitation in specialist care centres, often have tried other treatments for their chronic pain with no or little effect. That is, these patients have severe problems and relative treatment resistance. Interestingly, the changes in outcomes with moderate ES are broad and not limited to a single outcome domain, and the most stable moderate ES were demonstrated for pain intensity aspects with moderate improvement both immediately after IMMRPs and at 12-month follow-up. Pain interference demonstrated moderate ES improvement at 12-month follow-up, and vitality was moderately improved immediately after IMMRPs. Both objective registrations (e.g., sick-leave registrations and actigraphic recordings [54]) and subjective PROM data may be important for understanding the efficacy of IMMRPs. Very recently, a SQRP study using a subgroup of the same cohort of patients reported that sick leave benefits according to the Swedish Social Insurance Agency decreased as a consequence of IMMRP [55]. Hence, both PROM data and objective sick leave data indicate clinically important positive changes in response to IMMRPs for patients in real-life practice settings. As a comparison, SRs conclude that common pharmacological treatments—e.g., paracetamol, non-steroidal anti-inflammatory drugs, and opioids—for patients with chronic pain have no effects, small effects, and/or lack of long-term follow-up effects [56,57,58].

The present study reported medium ES for two of three pain intensity variables both post-IMMRP and at 12-month FU (i.e., for MPI-pain-severity and sf36-bp); the third pain intensity variable had effect sizes near medium ES. These results contrasted some SRs reporting of no evidence for efficacy with respect to pain intensity [10,11]. However, not all RCTs of IMMRP included pain intensity outcomes, since the interventions are not focused on the pain itself but rather on its consequences [10,11]. Obviously, many pain patients consider pain intensity improvement to be the most important aspect of treatments [59]. However, changing this perspective is considered important in IMMPRs, since focusing on pain reduction in many cases leads to short-sighted attempts to control pain, and this may, when not successful, lead to increased physical and psychological disability and reduced life quality [60,61]. Thus, specialist care IMMRPs in Sweden have largely adopted the idea of introducing acceptance as a cornerstone of the psychological component of IMMRP (i.e., the willingness to have the experiences of pain as it is and to encourage patients to set up activity-related rehabilitation goals and risk initial pain flare-ups). This also means that patients are advised against establishing pain reduction goals. Thus, it could be considered problematic to communicate the present results showing medium effect sizes in real-world practice settings on pain. On the other hand, it may also be ethically problematic if both clinical practice and research ignore the reports and wishes of the patients regarding pain intensity. However, health care providers should not underestimate their patients’ ability to grasp, once explained, the complex pain experience. Therefore, health care providers should emphasise pain education, including descriptions of the affective and cognitive elements of pain as rational for the different components of IMMRPs, and stress the need to experiment with new behaviours and risk short-term pain flare-ups. Since the results are obtained in this context, no change in clinical practise as far as pain communication is called for.

SRs of IMMRP report that it is an effective intervention with small to moderate effects for patients with chronic pain conditions [11,12,62,63]. The present results concerning ES agree with most SRs of IMMRP, but it may also be appropriate to compare with ES results reported in other clinical studies. The moderate ES for two of the pain intensity variables agree with studies in clinical routine care (n = 65–395), and therefore, for long-term follow-up (6–12 months), such studies report small (Cohen’s d: 0.20–0.33 [64,65]) to moderate (Cohen’s d: 0.59–0.70 [26,66,67]) ES for pain intensity. For psychological distress variables, these studies agree with the present results: they generally found small ES for long-term follow-up (i.e., Cohen’s d = 0–0.38 for depressive symptoms [26,64,65] and Cohen’s d = 0.22–0.34 for anxiety) [26,65]. In a recent RCT comparing transdiagnostic emotion-focused exposure treatment (Hybrid) and Internet-delivered pain management treatment (ICBT) for chronic pain patients with comorbid anxiety and depression, we found that within group ES pre versus follow-up for pain interference were reported both for hybrid (ES = 1.17) and for ICBT (ES = 0.65) compared the present effect size of 0.49 [68]. However, the patients were not exclusively recruited from specialist care (i.e., clinical departments of pain rehabilitation); they were also recruited via advertisements in local newspapers and social media [68], and the numbers investigated were considerably smaller. An important observation from the present study is that moderate ES found at 12-month follow-up covered broad aspects (e.g., pain intensity, interference in daily life, and perceived health).

The number of outcomes in IMMRPs in RCTs are generally high, which reasonably reflects the broad goals of the complex intervention. The present study used 22 outcomes that are mandatory in SQRP measured on up to three occasions (i.e., pre-IMMRP, post-IMMRP, and 12-month follow-up). PCA was applied to handle the pattern of changes in potentially intercorrelated outcomes as suggested by the Medical Research Council of the United Kingdom [69]. From these analyses, it can be concluded that changes in pain intensity, pain interference, psychological distress, vitality, etc. were positively intercorrelated (Table 5). In fact, our study showed that the changes in the majority of the 22 outcomes are significantly intercorrelated. Hence, the changes in these variables cannot be considered independent of each other. As a consequence of this observation, the appropriateness to evaluate changes in outcomes separately, as done in a recent SR [70], must be questioned, since the treatment was not designed to target only a single outcome. Moreover, the ES must be seen in this complex context. Thus, small changes in many outcomes may be more important than one prominent change in a single or few outcomes. Furthermore, the Grading of Recommendations Assessment, Development and Evaluation (GRADE) used for evidence ratings in SRs may not adequately describe the evidence base of complex interventions [71]. Different definitions of positive outcomes of IMMRP interventions exist (e.g., the majority of outcomes had to be significantly better than for the control intervention) [10,11]. Another approach was that the authors of the SR predetermined primary and secondary outcomes and what was necessary to classify an intervention as positive before reviewing the RCTs [12]. 

The presented PCAs also highlight that it may be possible to reduce the number of outcome variables, since several of these appear to measure similar aspects of the chronic pain condition. The fact that 22 outcomes were analysed (Table 2 and Table 3) may raise an issue of multiple comparisons. In such situations, Bonferroni corrections are frequently used [72,73]. This is a conservative approach when the number of tests increases [72,74,75], the chances to detect real treatment effects decrease, and corrections were designed for corrections of *independent* comparisons [74]. The latter is obviously not present for most changes in outcome variables according to the PCAs performed (Table 5). Hierarchal or ‘gatekeeping’ procedures do not require adjustment for multiplicity [73], but require a natural hierarchy of the outcomes, as such a hierarchy is not obvious for IMMRP, as discussed above. Another approach is that outcomes are combined into a single composite outcome (i.e., a composite outcome consists of two or more component outcomes) [76], but this may be problematic with respect to missing cases and when the components of the composite endpoint are measured on different scales (i.e., non-commensurate outcomes) [76]. However, some multivariate methods such as PCA and OPLS can handle non-commensurate outcomes [76]. We used advanced PCA, including the NIPALS algorithm, to handling missing data and non-commensurate outcomes. We calculated the *t*-scores for the most relevant latent factor (component). Hence, we defined an objective Multivariate Improvement Score (MIS; the *t*-score of the first PCA component), which on an individual patient level defines the multivariate improvement; a positive MIS indicates multivariate improvements because of IMMRP. 

Three clearly separated clusters based on MIS were identified. On a group level, clusters 1 and 2 were associated with various degrees of improvements, whereas cluster 3 showed negative MIS, indicating deterioration. Although the greater improvement in cluster 1 can be interpreted as a sign of regression to the mean and that these patients did not benefit from IMMRP more than cluster 2, this cluster still improves from IMMRP at least as well as those with e.g., less severe psychological distress symptoms (clusters 2 and 3). This may seem unexpected, but it is important to recognise that addressing psychological symptoms with CBT is an important component of IMMRPs. The patients at post-IMMRP and 12-month follow-up estimated the degree of positive change in pain (i.e., Change-pain) and the ability to handle life situation in general (i.e., Change-life situation). Most patients reported improvements according to both the Change-pain and Change-life situations (Table 4). Relatively small proportions of the patients reported worse situations post-IMMRP and at the 12-month follow-up, which are results that agree with other studies [29,77,78]. These two variables have retrospective elements even though they are not explicitly expressed. There are several problems with such items in general—e.g., desirability and memory aspects, recall time [79,80,81], and in treatment context (e.g., overly optimistic assessments) [82]. However, on a general level, these estimations and the two MIS variables (Table 6 and Table 7) agreed.

We found that cluster 1, which had high MIS values (i.e., prominent improvements), had a more severe clinical picture at baseline/pre-IMMRP than those with lower MIS (i.e., less improvements). These results agree with another SQRP study (*N* > 35,000) that identified clusters based on the clinical presentation at assessment (decision not taken about participation in IMMRP); the study found that patients with the most severe clinical situation who later participated in IMMRP had the most prominent improvements in six investigated outcomes [34]. Although IMMRP has been commended for its effectiveness (‘of all approaches to the treatment of chronic pain, none has a stronger evidence basis for efficacy, cost-effectiveness, and lack of iatrogenic complications’) [83], both this and our recent study [34] indicate that not all patients show important improvements in several domains of outcome after IMMRP. Both this and our previous study identified a large subgroup of patients that do not seem to significantly benefit from IMMRP. Presumably, these patients—in the present study, those with negative MIS (i.e., cluster 3)—need other interventions. In a relative context, they have a somewhat less complicated self-reported clinical picture pre-IMMRP than those in clusters 1 and 2, even though they are referred to specialist care and hence represent patients with complex needs. 

The longitudinal regressions of MIS using background variables and pre-IMMRP data as regressors were significant (Table 10). A blend of variables was important; psychological distress variables were most important, but life impact variables, pain aspects, and health and vitality aspects contributed. Our results appear to be in line with a recent meta-analysis on prognostic factors for IMMRP outcome, demonstrating that both pre-treatment general emotional distress and pain-specific cognitive behavioural factors are related to worse long-term (>6 months) physical functioning [84]. Unfortunately, these regressions cannot be used clinically, since they only explained 8% of variations in MIS. Although the prediction does not work clinically, this and a previous study from our group give clear indications that patients with a severe clinical situation benefit from IMMRP [34].

### 4.1. Important Clinical Implications

Outcomes of IMMRP in real-life practice settings agree with the conclusions from SRs. Partly in contrast to SRs, this registry study of patients managed within specialist care found that pain intensity was positively affected because of IMMRP. It was also obvious that not all patients benefit from IMMRP. Hence, there is a need to develop better matching between clinical presentation and participation in MMRP in real-life practice settings. Moreover, the intercorrelations of most changes in outcomes also opens up the possibility of reducing the number of outcome variables and hereby reduce the burden upon patients included in the SQRP. 

### 4.2. Strengths and Limitations

This study’s strengths include a large number of patients with complex chronic pain conditions with a nation-wide representation. However, these patients were referred to specialist clinics and thus represent a selection of the most difficult cases, so our results cannot be generalised to other settings. Another strength was the use of MVDA methods such as PCA and OPLS to handle correlation patterns, repeated measures, and regressions when there were obvious risks for multicollinearity. Changes in the social context may have changed and influenced the longitudinal analyses; however, we used validated and well-known instruments. Repeated evaluations using PROM questionnaires in treatment studies may be problematic [85]. Thus, the changes that the patient undergo because of the intervention (i.e., IMMRP) may affect the interpretations of the questions when presented at follow-up. The fact that no control group or treatment-as-usual group was available, which ethically is complicated to arrange for a registry of real-life practice patients, might have influenced our interpretation of changes after IMMRP. Data for the time period 2008–2016 from the SQRP was used in the present study, and changes in the content of IMMRP may have occurred. Unfortunately, no data concerning such changes are available.

## 5. Conclusions

This large-scale study of IMMRPs in real life practise settings demonstrates significant outcome changes in almost all measures. Most short-term and long-term effect sizes were small, but interestingly, moderate long-term effect sizes were demonstrated for pain, pain interference in daily life, and perceived health. In addition, patients reporting higher levels of perceived disability and suffering displayed greater improvement. 

## Figures and Tables

**Figure 1 jcm-08-00905-f001:**
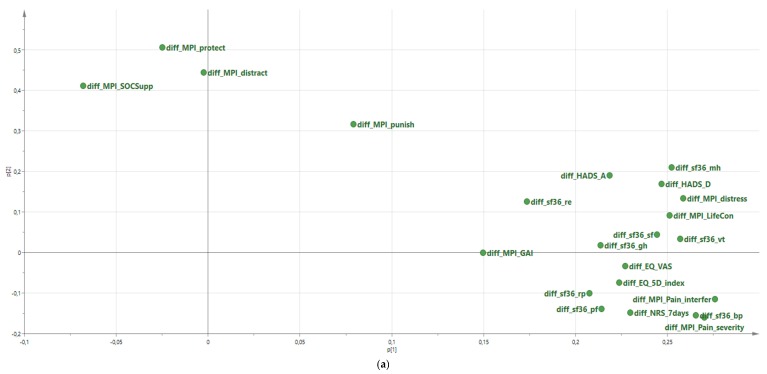
Loading plot of changes (pre-IMMRP vs. 12-month follow-up) in the 22 outcome variables—i.e., the relationships between the changes (**a**) and score plot ((**b**); the relationships between the patients). diff = change in a certain variable; NRS-7days = Pain intensity as measured by a numeric rating scale for the previous seven days; HADS = Hospital Anxiety and Depression Scale; MPI = Multidimensional Pain Inventory; EQ-5D-index = The index of the European quality of life instrument; EQ-VAS = The European quality of life instrument thermometer-like scale; sf36 = The Short Form (36) Health Survey; For explanations of the subscale abbreviations, see Methods.

**Table 1 jcm-08-00905-t001:** Continuous background variables; mean ± SD and 95% confidence intervals (95% CI).

Variables	Mean ± SD	95% CI Lower Bound	95% CI Upper Bound
Age (years)	43.2 ± 10.7	43.3	43.9
Days no work	1055 ± 2461	968	1095
Pain duration	3057 ± 3341	2970	3170
Persistent pain duration	2368 ± 2980	2239	2414
PRI	15.4 ± 8.6	15.1	15.6

Notes: SD = standard deviation; CI = confidence intervals; PRI = Pain Region Index.

**Table 2 jcm-08-00905-t002:** Outcome variables at baseline (pre-IMMRP) and immediately after IMMRP (post-IMMRP). Statistical comparisons are presented furthest to the right together with effects sizes (i.e., Cohen’s d). Effect sizes in bold were moderate, i.e., Cohen’s d ≥ 0.50. IMMRP: interdisciplinary multimodal pain rehabilitation programs.

Baseline vs. After IMMRP		Pre-IMMRP		Post-IMMRP			
	*N*	Mean	SD	Mean	SD	*p*-Value	Cohen’s d
NRS-7days	14,146	6.86	1.72	5.95	2.09	<0.001	0.45
HADS-A	14,774	9.00	4.76	7.78	4.55	<0.001	0.32
HADS-D	14,772	8.49	4.44	6.70	4.31	<0.001	0.47
MPI-Pain-severity	14,692	4.39	0.93	3.87	1.16	<0.001	**0.52**
MPI-Pain-interfer	14,552	4.38	1.02	3.94	1.19	<0.001	0.49
MPI-LifeCon	14,687	2.72	1.10	3.30	1.18	<0.001	0.47
MPI-Distress	14,697	3.46	1.26	2.89	1.38	<0.001	0.42
MPI-Socsupp	14,618	4.16	1.34	3.95	1.35	<0.001	0.21
MPI-punish	13,054	1.74	1.36	1.72	1.33	0.037	0.02
MPI-protect	12,999	2.98	1.40	2.85	1.38	<0.001	0.12
MPI-distract	13,048	2.54	1.19	2.56	1.17	0.043	0.02
MPI-GAI	14,676	2.44	0.84	2.63	0.82	<0.001	0.26
EQ-5D-index	13,989	0.26	0.31	0.39	0.33	<0.001	0.40
EQ-VAS	13,777	41.22	19.09	50.99	21.38	<0.001	0.44
sf36-pf	14,253	52.76	20.58	57.67	21.17	<0.001	0.30
sf36-rp	13,945	12.53	24.40	22.46	33.12	<0.001	0.30
sf36-bp	14,268	24.36	14.49	32.96	17.41	<0.001	**0.52**
sf36-gh	13,988	41.70	20.22	46.69	21.88	<0.001	0.29
sf36-vt	14,206	23.95	18.48	35.67	22.76	<0.001	**0.54**
sf36-sf	14,229	47.29	25.19	54.93	25.91	<0.001	0.30
sf36-re	13,701	42.77	42.92	51.15	43.48	<0.001	0.18
sf36-mh	14,194	55.03	21.35	62.55	21.55	<0.001	0.38

NRS-7days = Pain intensity as measured by a numeric rating scale for the previous seven days; HADS = Hospital Anxiety and Depression Scale; MPI = Multidimensional Pain Inventory; EQ-5D-index = The index of the European quality of life instrument; EQ-VAS = The European quality of life instrument thermometer-like scale; sf36 = The Short Form (36) Health Survey. For explanations of the subscale abbreviations, see Methods.

**Table 3 jcm-08-00905-t003:** Outcome variables at baseline (pre-IMMRP) and at the 12-month follow-up (FU). Statistical comparisons are presented furthest to the right together with effects sizes (i.e., Cohen’s d). Effect sizes in bold were moderate (i.e., Cohen’s d ≥ 0.50).

Baseline vs. 12-Month Follow-Up		Pre IMMRP		12-Month FU			
	*N*	Mean	SD	Mean	SD	*p*-Value	Cohen’s d
NRS-7days	8568	6.84	1.72	5.78	2.32	<0.001	0.47
HADS-A	8865	8.73	4.69	7.38	4.70	<0.001	0.33
HADS-D	8865	8.18	4.37	6.74	4.66	<0.001	0.35
MPI-Pain-severity	8904	4.36	0.91	3.71	1.33	<0.001	**0.56**
MPI-Pain-interfer	8829	4.34	1.02	3.73	1.37	<0.001	**0.54**
MPI-LifeCon	8871	2.77	1.10	3.28	1.27	<0.001	0.40
MPI-Distress	8889	3.42	1.27	2.92	1.45	<0.001	0.35
MPI-Socsupp	8830	4.17	1.33	3.77	1.42	<0.001	0.35
MPI-punish	7824	1.69	1.34	1.69	1.35	0.676	0.01
MPI-protect	7784	2.96	1.39	2.78	1.40	<0.001	0.16
MPI-distract	7811	2.52	1.17	2.45	1.17	<0.001	0.06
MPI-GAI	8859	2.47	0.83	2.64	0.86	<0.001	0.20
EQ-5D-index	8844	0.27	0.31	0.44	0.34	<0.001	**0.50**
EQ-VAS	8607	41.90	19.29	52.96	22.87	<0.001	0.46
sf36-pf	8459	53.07	20.30	59.73	22.57	<0.001	0.36
sf36-rp	8301	13.07	24.91	27.74	36.32	<0.001	0.39
sf36-bp	8458	24.60	14.11	35.41	20.05	<0.001	**0.56**
sf36-gh	8342	42.59	20.49	47.35	23.52	<0.001	0.25
sf36-vt	8441	24.96	18.79	34.41	23.85	<0.001	0.41
sf36-sf	8459	48.95	25.50	57.66	27.05	<0.001	0.32
sf36-re	8159	44.69	43.17	55.60	43.53	<0.001	0.22
sf36-mh	8435	56.34	21.15	62.70	22.53	<0.001	0.30

NRS-7days = Pain intensity as measured by a numeric rating scale for the previous seven days; HADS = Hospital Anxiety and Depression Scale; MPI = Multidimensional Pain Inventory; EQ-5D-index = The index of the European quality of life instrument; EQ-VAS = The European quality of life instrument thermometer-like scale; sf36 = The Short Form (36); Health Survey; FU = Follow-up. For explanations of the subscale abbreviations see Methods.

**Table 4 jcm-08-00905-t004:** Estimations of pain situation (Change-pain) and in the ability to handle life situation in general (Change-life situation) made immediately after IMMRP (post-IMMRP) and at the 12-month FU.

Change-Pain		Post-IMMRP		12-Month FU
	*n*	%	*n*	%
0. Markedly increased pain	447	3.2	225	2.6
1. Partially increased pain	1517	11	590	6.9
2. No change	4008	29.1	2905	34
3. Partially diminished pain	6178	44.9	3662	42.8
4. Markedly diminished pain	1607	11.7	1174	13.7
Total	13 757	100	8 556	100
**Change-Life situation**		**Post-IMMRP**		**12-Month FU**
	***n***	**%**	***n***	**%**
0. Markedly deteriorated	74	0.5	108	1.3
1. Partially deteriorated	248	1.8	282	3.3
2. No change	1923	13.9	1615	18.8
3. Partially improved	8412	60.9	4628	54
4. Markedly improved	3161	22.9	1937	22.6
Total	13 818	100	8 570	100

FU = Follow-up.

**Table 5 jcm-08-00905-t005:** Principal component analysis (PCA) of changes pre-IMMRP vs. post-IMMRP (left part) and pre-IMMRP vs. 12-month FU (right part). The significant components (p) are shown. Absolute loadings ≥ 0.25 are shown in bold to facilitate interpretation. Changes in outcomes are calculated so that a positive value indicates an improvement.

	Changes Pre-IMMRP vs. Post-IMMRP	Changes Pre-IMMRP vs. 12-Month FU		
	p[1]	p[1]	p[2]	p[3]
diff-NRS-7days	0.23	0.23	−0.15	**0.29**
diff-HADS-A	0.23	0.22	0.19	**−0.33**
diff-HADS-D	**0.26**	**0.25**	0.17	−0.24
diff-MPI-Pain-sever	**0.27**	**0.27**	−0.16	0.26
diff-MPI-Pain-interfer	**0.26**	**0.28**	−0.11	0.10
diff-MPI-LifeCon	**0.26**	**0.25**	0.09	−0.05
diff-MPI-distress	**0.27**	**0.26**	0.13	−0.21
diff-MPI-SOCsupp	−0.03	−0.07	**0.41**	0.21
diff-MPI-punish	0.07	0.08	**0.32**	0.11
diff-MPI-protect	−0.02	−0.02	**0.51**	**0.33**
diff-MPI-distract	0.01	0.00	**0.45**	**0.36**
diff-MPI-GAI	0.12	0.15	0.00	0.07
diff-EQ-5D-index	0.22	0.22	−0.07	0.12
diff-EQ-VAS	0.22	0.23	−0.03	0.09
diff-sf36-pf	0.20	0.21	−0.14	0.20
diff-sf36-rp	0.19	0.21	−0.10	0.17
diff-sf36-bp	**0.26**	**0.27**	−0.15	**0.25**
diff-sf36-gh	0.21	0.21	0.02	0.02
diff-sf36-vt	**0.27**	**0.26**	0.03	−0.03
diff-sf36-sf	**0.25**	0.24	0.05	−0.09
diff-sf36-re	0.18	0.17	0.13	**−0.26**
diff-sf36-mh	**0.27**	**0.25**	0.21	**−0.30**
R^2^	0.31	0.36	0.10	0.06
Q^2^	0.25	0.31	0.04	0.02
N	14,666	8851		

diff = change in a certain variable; *p* = principal component; NRS-7days = Pain intensity as measured by a numeric rating scale for the previous seven days; HADS = Hospital Anxiety and Depression Scale; MPI = Multidimensional Pain Inventory; EQ-5D-index = The index of the European quality of life instrument; EQ-VAS = The European quality of life instrument thermometer-like scale; sf36 = The Short Form (36) Health Survey; FU = Follow-up. For explanations of the subscale abbreviations see Methods.

**Table 6 jcm-08-00905-t006:** Clusters from hierarchical clustering analysis (HCA) based on Multivariate Improvement Score (MIS) (*t*-scores) of the first component of the PCA for the changes in outcomes from pre-IMMRP to post-IMMRP (denoted ***MIS post-IMMRP***). To facilitate understanding, the changes for all the outcomes are shown (mean, SD, and 95% confidence interval). The two bottom rows show the estimations of changes (not included in PCA and the calculation of MIS).

	Cluster 1 (15.0%)					Cluster 2 (54.1%)					Cluster 3 (30.8%)						
	*N*	Mean	SD	95% CI		*N*	Mean	SD	95% CI		*N*	Mean	SD	95% CI		ANOVA	
Variables				LB	UB				LB	UB				LB	UB	*p*-Value	Post Hoc
***MIS post-IMMRP***	***2205***	***4.37***	***1.61***	***4.31***	***4.44***	***7938***	***0.33***	***1.08***	***0.30***	***0.35***	***4523***	***−2.74***	***1.16***	***−2.78***	***−2.71***	***<0.001***	***all different***
diff-NRS-7days	2086	3.04	2.05	2.95	3.13	7511	1.01	1.69	0.97	1.04	4267	−0.28	1.58	−0.32	−0.23	<0.001	all different
diff-HADS-A	2191	5.12	3.72	4.96	5.28	7873	1.59	3.08	1.52	1.66	4458	−1.30	3.24	−1.39	−1.20	<0.001	all different
diff-HADS-D	2188	5.93	3.55	5.78	6.08	7869	2.24	2.95	2.17	2.30	4462	−0.96	3.04	−1.05	−0.87	<0.001	all different
diff-MPI-Pain-severity	2185	1.77	1.02	1.73	1.82	7874	0.56	0.74	0.54	0.58	4500	−0.17	0.69	−0.19	−0.15	<0.001	all different
diff-MPI-Pain-interfer	2173	1.53	0.99	1.49	1.57	7811	0.50	0.70	0.48	0.51	4453	−0.17	0.66	−0.18	−0.15	<0.001	all different
diff-MPI-LideCon	2189	1.94	1.05	1.89	1.98	7861	0.71	0.96	0.69	0.73	4491	−0.33	0.99	−0.36	−0.30	<0.001	all different
diff-MPI-distress	2187	2.18	1.17	2.13	2.23	7869	0.71	1.02	0.69	0.73	4487	−0.46	1.05	−0.49	−0.43	<0.001	all different
diff-MPI-SOCsupp	2180	−0.34	1.12	−0.39	−0.30	7828	−0.24	0.98	−0.26	−0.21	4462	−0.11	0.98	−0.14	−0.08	<0.001	all different
diff-MPI-punish	1991	0.39	1.20	0.34	0.45	7016	0.04	1.12	0.02	0.07	4008	−0.20	1.15	−0.24	−0.17	<0.001	all different
diff-MPI-protect	1988	−0.18	1.17	−0.23	−0.13	6984	−0.14	1.00	−0.17	−0.12	3989	−0.05	1.05	−0.09	−0.02	<0.001	cl1 NE cl2, cl2 NE cl3
diff-MPI-distract	1992	0.11	1.10	0.06	0.16	7012	0.00	0.96	−0.03	0.02	4006	0.01	1.03	−0.02	0.04	<0.001	cl1 NE cl2, cl3
diff-MPI-GAI	2187	0.57	0.82	0.53	0.60	7866	0.23	0.68	0.21	0.24	4489	−0.06	0.69	−0.08	−0.03	<0.001	all different
diff-EQ-5D-index	2105	0.44	0.30	0.43	0.45	7494	0.16	0.28	0.15	0.17	4205	−0.07	0.28	−0.08	−0.06	<0.001	all different
diff-EQ-VAS	2068	30.51	19.66	29.66	31.36	7409	11.61	18.58	11.19	12.03	4126	−3.84	19.15	−4.42	−3.25	<0.001	all different
diff-sf36-pf	2147	18.74	16.97	18.02	19.46	7644	6.02	13.63	5.72	6.33	4324	−3.83	14.22	−4.25	−3.40	<0.001	all different
diff-sf36-rp	2110	39.93	39.28	38.25	41.60	7513	10.46	29.70	9.79	11.13	4215	−5.96	25.56	−6.74	−5.19	<0.001	all different
diff-sf36-bp	2146	28.04	16.27	27.35	28.73	7663	9.72	12.96	9.43	10.01	4323	−2.93	12.38	−3.30	−2.56	<0.001	all different
diff-sf36-gh	2126	21.06	17.59	20.31	21.81	7529	6.05	14.51	5.72	6.37	4235	−4.89	14.57	−5.33	−4.45	<0.001	all different
diff-sf36-vt	2139	37.45	18.74	36.65	38.24	7626	13.58	17.26	13.19	13.96	4311	−4.20	16.16	−4.69	−3.72	<0.001	all different
diff-sf36-sf	2146	34.26	22.17	33.32	35.20	7652	10.12	20.39	9.66	10.58	4324	−9.86	21.10	−10.49	−9.23	<0.001	all different
diff-sf36-re	2087	43.52	44.58	41.60	45.43	7390	11.98	43.21	11.00	12.97	4121	−15.81	42.02	−17.09	−14.52	<0.001	all different
diff-sf36-mh	2139	29.61	17.48	28.86	30.35	7620	10.12	14.90	9.78	10.45	4307	−8.01	15.89	−8.49	−7.54	<0.001	all different
Change-Pain	2059	3.28	0.69	3.25	3.31	7280	2.53	0.87	2.51	2.55	4015	2.08	0.93	2.05	2.11	<0.001	all different
Change-Life situation	2067	3.51	0.58	3.48	3.53	7315	3.07	0.63	3.05	3.08	4032	2.76	0.72	2.74	2.79	<0.001	all different

LB = Lower Bound; UB = Upper Bound; diff = change in a certain variable; NRS-7days = Pain intensity as measured by a numeric rating scale for the previous seven days; HADS = Hospital Anxiety and Depression Scale; MPI = Multidimensional Pain Inventory; EQ-5D-index = The index of the European quality of life instrument; EQ-VAS = The European quality of life instrument thermometer-like scale; sf36 = The Short Form (36) Health Survey. For explanations of the subscale abbreviations, see Methods.

**Table 7 jcm-08-00905-t007:** Clusters from HCA based on MIS (*t*-scores) of the first component of the PCA for the changes in outcomes from pre-IMMRP to 12-month FU (denoted as ***MIS 12-m FU***). To facilitate understanding, the changes for all the outcomes are shown (mean, SD, and 95% confidence interval). The two bottom rows show the estimations of changes (not included in PCA and the calculation of MIS).

	Cluster 1 (12.4%)					Cluster 2 (46.6%)					Cluster 3 (41.0%)						
	*N*	Mean	SD	95% CI		*N*	Mean	SD	95% CI		*N*	Mean	SD	95% CI		ANOVA	
Variables				LB	UB				LB	UB				LB	UB	*p*-Value	Post Hoc
***MIS -12-m FU***	***1099***	***5.01***	***1.78***	***4.90***	***5.11***	***4123***	***0.78***	***1.35***	***0.74***	***0.82***	***3629***	***−2.43***	***1.39***	***−2.47***	***−2.38***	***<0.001***	***all different***
diff-NRS-7days	1031	3.67	2.15	3.54	3.80	3876	1.46	1.89	1.40	1.52	3435	−0.16	1.66	−0.21	−0.10	<0.001	all different
diff-HADS-A	1095	5.72	3.84	5.49	5.95	4087	2.06	3.34	1.96	2.16	3588	−0.80	3.53	−0.91	−0.68	<0.001	all different
diff-HADS-D	1096	6.13	3.75	5.90	6.35	4086	2.39	3.13	2.29	2.48	3588	−1.05	3.39	−1.16	−0.94	<0.001	all different
diff-MPI-Pain-severity	1092	2.30	1.14	2.24	2.37	4108	0.88	0.90	0.85	0.91	3619	−0.09	0.76	−0.12	−0.07	<0.001	all different
diff-MPI-Pain-interfer	1092	2.30	1.14	2.24	2.37	4108	0.88	0.90	0.85	0.91	3589	−0.14	0.74	−0.17	−0.12	<0.001	all different
diff-MPI-LifeCon	1086	2.27	1.10	2.21	2.34	4076	0.83	0.85	0.81	0.86	3604	−0.25	1.05	−0.28	−0.21	<0.001	all different
diff-MPI-distress	1091	2.10	1.10	2.03	2.16	4095	0.75	1.00	0.71	0.78	3618	−0.38	1.13	−0.41	−0.34	<0.001	all different
diff-MPI-SOCsupp	1086	−0.69	1.25	−0.77	−0.62	4081	−0.60	1.16	−0.64	−0.57	3583	−0.10	1.04	−0.13	−0.07	<0.001	all different
diff-MPI-punish	979	0.57	1.19	0.50	0.65	3633	−0.02	1.22	−0.06	0.02	3194	−0.13	1.22	−0.18	−0.09	<0.001	all different
diff-MPI-protect	980	−0.23	1.24	−0.31	−0.15	3616	−0.35	1.20	−0.39	−0.31	3173	0.03	1.09	−0.01	0.06	<0.001	all different
diff-MPI-distract	981	0.02	1.15	−0.05	0.10	3627	−0.22	1.09	−0.25	−0.18	3188	0.08	1.02	0.04	0.11	<0.001	CL NE cl3, cl2 NE cl3
diff-MPI-GAI	1090	0.76	0.95	0.70	0.82	4094	0.26	0.73	0.23	0.28	3613	−0.11	0.73	−0.14	−0.09	<0.001	all different
diff-EQ-5D-index	1048	0.53	0.30	0.51	0.55	3886	0.25	0.30	0.24	0.26	3351	−0.04	0.30	−0.05	−0.03	<0.001	all different
diff-EQ-VAS	1022	36.06	20.47	34.80	37.32	3833	16.27	19.82	15.64	16.89	3316	−3.31	19.66	−3.98	−2.64	<0.001	all different
diff-sf36-pf	1054	26.77	19.09	25.61	27.92	3953	10.00	15.13	9.53	10.47	3445	−3.31	15.24	−3.81	−2.80	<0.001	all different
diff-sf36-rp	1043	57.34	39.51	54.94	59.74	3889	19.15	34.30	18.07	20.22	3365	−3.69	27.14	−4.60	−2.77	<0.001	all different
diff-sf36-bp	1054	37.44	19.56	36.26	38.63	3954	14.29	14.79	13.83	14.75	3444	−1.33	13.03	−1.76	−0.89	<0.001	all different
diff-sf36-gh	1044	25.26	19.09	24.10	26.42	3908	7.78	16.33	7.26	8.29	3388	−5.04	15.87	−5.57	−4.50	<0.001	all different
diff-sf36-vt	1054	40.71	19.72	39.52	41.91	3951	13.32	17.83	12.77	13.88	3433	−4.60	16.67	−5.16	−4.05	<0.001	all different
diff-sf36-sf	1053	40.73	23.26	39.32	42.14	3958	14.43	21.21	13.77	15.09	3445	−7.66	22.38	−8.40	−6.91	<0.001	all different
diff-sf36-re	1031	53.10	44.27	50.40	55.81	3849	17.22	45.08	15.79	18.64	3277	−9.79	45.59	11.35	−8.22	<0.001	all different
diff-sf36-mh	1054	32.77	18.81	31.64	33.91	3948	10.62	16.33	10.11	11.13	3430	−6.66	17.25	−7.24	−6.09	<0.001	all different
Change-Pain	1049	3.32	0.74	3.28	3.37	3887	2.72	0.81	2.69	2.75	3373	2.20	0.85	2.17	2.22	<0.001	all different
Change-Life situation	1049	3.52	0.64	3.48	3.56	3901	3.05	0.70	3.03	3.08	3375	2.62	0.83	2.59	2.65	<0.001	all different

LB = Lower Bound; UB = Upper Bound; diff = change in a certain variable; NRS-7days = Pain intensity as measured by a numeric rating scale for the previous seven days; HADS = Hospital Anxiety and Depression Scale; MPI = Multidimensional Pain Inventory; EQ-5D-index = The index of the European quality of life instrument; EQ-VAS = The European quality of life instrument thermometer-like scale; sf36 = The Short Form (36) Health Survey. For explanations of the subscale abbreviations, see Methods.

**Table 8 jcm-08-00905-t008:** Pre-IMMRP values for the three clusters based on MIS obtained post-IMMRP.

	Cluster 1					Cluster 2					Cluster 3						
Baseline	*N*	Mean	SD	95% CI		*N*	Mean	SD	95% CI		*N*	Mean	SD	95% CI		ANOVA	
Variables				LB	UB				LB	UB				LB	UB	*p*-Value	Post Hoc
Gender	2205	0.22	0.41	0.20	0.23	7938	0.24	0.43	0.23	0.25	4523	0.25	0.43	0.24	0.26	0.014	NA
Age	2205	42.7	11.2	42.2	43.2	7938	43.5	10.7	43.3	43.8	4523	43.0	10.6	42.7	43.3	0.001	cl1 NE cl3, cl2 NE cl3
Outside-Europe	2185	0.11	0.31	0.09	0.12	7877	0.10	0.30	0.09	0.10	4484	0.11	0.32	0.10	0.12	0.011	NA
University	2168	0.28	0.45	0.26	0.29	7826	0.26	0.44	0.25	0.27	4445	0.23	0.42	0.22	0.24	0.000	cl1 NE cl2, cl2 NE cl3
Days no work	716	889	2912	675	1102	2976	1037	2311	954	1120	1774	1152	2502	1036	1269	0.045	NA
PRI	2205	13.8	8.3	13.4	14.1	7938	14.4	8.3	14.2	14.6	4523	14.5	8.3	14.3	14.7	0.002	NA
NRS-7days	2158	7.1	1.7	7.0	7.2	7801	6.9	1.7	6.8	6.9	4440	6.7	1.7	6.7	6.8	0.000	all different
HADS-A	2199	10.4	4.7	10.2	10.6	7891	9.0	4.7	8.9	9.1	4494	8.2	4.6	8.1	8.4	0.000	all different
HADS-D	2197	9.6	4.4	9.4	9.8	7892	8.6	4.4	8.5	8.7	4494	7.7	4.3	7.6	7.9	0.000	all different
MPI-Pain-severity	2193	4.5	0.9	4.5	4.6	7895	4.4	0.9	4.4	4.4	4510	4.3	0.9	4.3	4.3	0.000	all different
MPI-Pain-interfer	2187	4.6	1.0	4.6	4.6	7853	4.4	1.0	4.4	4.4	4485	4.3	1.0	4.2	4.3	0.000	all different
MPI-LifeCon	2195	2.4	1.1	2.4	2.5	7880	2.7	1.1	2.7	2.7	4502	2.9	1.1	2.9	3.0	0.000	all different
MPI-Distress	2191	3.9	1.2	3.9	4.0	7891	3.5	1.3	3.5	3.5	4502	3.2	1.3	3.1	3.2	0.000	all different
MPI-Socsupp	2190	4.2	1.4	4.2	4.3	7861	4.2	1.3	4.1	4.2	4488	4.1	1.4	4.1	4.2	0.005	NA
MPI-punish	2069	1.9	1.5	1.8	1.9	7319	1.8	1.4	1.7	1.8	4169	1.7	1.3	1.6	1.7	0.000	all different
MPI-protect	2065	3.0	1.5	2.9	3.1	7295	2.9	1.4	2.9	3.0	4158	3.0	1.4	2.9	3.0	0.136	NA
MPI-distract	2069	2.6	1.2	2.5	2.6	7312	2.5	1.2	2.5	2.5	4168	2.5	1.2	2.5	2.6	0.075	NA
MPI-GAI	2193	2.4	0.8	2.4	2.4	7885	2.4	0.8	2.4	2.5	4500	2.5	0.8	2.5	2.5	0.000	cl1 NE cl2, cl2 NE cl3
EQ-5D-index	2126	0.2	0.3	0.2	0.2	7587	0.3	0.3	0.2	0.3	4277	0.3	0.3	0.3	0.3	0.000	all different
EQ-VAS	2097	38.0	18.2	37.2	38.8	7529	40.7	19.0	40.3	41.1	4224	43.9	19.4	43.3	44.5	0.000	all different
sf36-pf	2151	52.1	21.0	51.2	52.9	7683	52.8	20.6	52.3	53.2	4349	53.2	20.2	52.6	53.8	0.119	NA
sf36-rp	2139	9.0	19.8	8.2	9.9	7629	11.8	23.8	11.3	12.4	4283	15.3	26.8	14.5	16.1	0.000	all different
sf36-bp	2151	21.1	13.8	20.5	21.7	7694	24.1	14.3	23.8	24.4	4347	26.4	14.6	26.0	26.8	0.000	all different
sf36-gh	2139	39.8	20.0	38.9	40.6	7610	41.4	20.1	41.0	41.9	4299	43.3	20.4	42.7	43.9	0.000	all different
sf36-vt	2151	19.7	16.6	19.0	20.4	7679	23.3	18.4	22.9	23.7	4347	27.3	19.0	26.7	27.8	0.000	all different
sf36-sf	2151	40.0	23.7	39.0	41.0	7690	46.6	24.9	46.1	47.2	4348	52.1	25.3	51.4	52.9	0.000	all different
sf36-re	2126	30.6	39.6	28.9	32.2	7549	41.6	42.6	40.7	42.6	4220	50.6	43.6	49.3	51.9	0.000	all different
sf36-mh	2151	47.3	20.7	46.5	48.2	7672	54.5	21.2	54.0	54.9	4343	60.1	20.5	59.4	60.7	0.000	all different

LB = Lower Bound; UB = Upper Bound; NA = not applicable; NRS-7days = Pain intensity as measured by a numeric rating scale for the previous seven days; HADS = Hospital Anxiety and Depression Scale; MPI = Multidimensional Pain Inventory; EQ-5D-index = The index of the European quality of life instrument; EQ-VAS = The European quality of life instrument thermometer-like scale; sf36 = The Short Form (36) Health Survey; PRI = Pain Region Index. For explanations of the subscale abbreviations, see Methods.

**Table 9 jcm-08-00905-t009:** Pre-IMMRP values for the three clusters based on MIS obtained at 12-month FU.

	Cluster 1					Cluster 2					Cluster 3						
Baseline	*N*	Mean	SD	95% CI		*N*	Mean	SD	95% CI		*N*	Mean	SD	95% CI		ANOVA	
Variables				LB	UB				LB	UB				LB	UB	*p*-Value	Post Hoc
Gender	1099	0.24	0.43	0.21	0.26	4123	0.21	0.41	0.20	0.22	3629	0.25	0.43	0.24	0.27	<0.001	all different
Age	1099	41.9	11.1	41.2	42.5	4123	43.7	11.1	43.4	44.0	3629	44.2	10.4	43.8	44.5	<0.001	all different
Outside-Europe	1088	0.09	0.29	0.07	0.11	4091	0.08	0.28	0.08	0.09	3606	0.10	0.30	0.09	0.11	0.012	NA
University	1079	0.29	0.45	0.26	0.31	4060	0.27	0.45	0.26	0.29	3567	0.23	0.42	0.21	0.24	<0.001	all different except cl2 vs. cl3
Days no work	358	661	2587	392	930	1409	1046	2339	923	1168	1323	1270	2604	1130	1411	<0.001	all different
PRI	1099	12.8	8.0	12.3	13.2	4123	14.2	8.3	14.0	14.5	3629	14.8	8.3	14.5	15.1	<0.001	all different
NRS-7days	1071	7.0	1.7	6.9	7.1	4049	6.8	1.7	6.8	6.9	3574	6.8	1.7	6.8	6.9	0.049	NA
HADS-A	1096	9.7	4.7	9.4	10.0	4101	8.7	4.7	8.6	8.9	3606	8.4	4.6	8.3	8.6	<0.001	all different
HADS-D	1096	8.8	4.4	8.6	9.1	4102	8.2	4.3	8.1	8.3	3606	8.0	4.4	7.9	8.1	<0.001	all different
MPI-Pain-severity	1094	4.5	0.9	4.4	4.5	4114	4.4	0.9	4.3	4.4	3625	4.3	0.9	4.3	4.4	<0.001	cl1NE cl2, cl2 NE cl1, cl3
MPI-Pain-interfer	1093	4.5	0.9	4.5	4.6	4101	4.4	1.0	4.3	4.4	3609	4.3	1.0	4.2	4.3	<0.001	all different
MPI-LifeCon	1096	2.5	1.1	2.5	2.6	4113	2.8	1.1	2.7	2.8	3614	2.9	1.1	2.8	2.9	<0.001	all different
MPI-Distress	1095	3.8	1.2	3.8	3.9	4114	3.4	1.3	3.4	3.5	3621	3.3	1.3	3.2	3.3	<0.001	all different
MPI-Socsupp	1090	4.3	1.3	4.2	4.4	4097	4.2	1.3	4.2	4.3	3604	4.1	1.4	4.1	4.1	<0.001	all different
MPI-punish	1036	1.8	1.4	1.7	1.9	3833	1.7	1.3	1.6	1.7	3355	1.7	1.3	1.7	1.8	0.047	NA
IMP-protect	1038	3.0	1.4	2.9	3.1	3824	2.9	1.4	2.9	3.0	3341	2.9	1.4	2.9	3.0	0.583	NA
MPI-distract	1039	2.6	1.2	2.5	2.6	3831	2.6	1.2	2.5	2.6	3352	2.5	1.2	2.4	2.5	0.003	NA
MPI -GAI	1094	2.4	0.9	2.4	2.5	4108	2.5	0.8	2.5	2.5	3623	2.5	0.8	2.5	2.5	0.071	NA
EQ-5D-index	1058	0.2	0.3	0.2	0.2	3935	0.3	0.3	0.3	0.3	3407	0.3	0.3	0.3	0.3	<0.001	all different
EQ-VAS	1037	39.2	18.2	38.1	40.3	3905	41.3	19.2	40.7	42.0	3389	43.1	19.6	42.4	43.7	<0.001	all different
sf36-pf	1056	51.9	20.4	50.7	53.1	3969	53.0	20.4	52.4	53.7	3459	53.4	20.1	52.7	54.1	0.120	NA
sf36-rp	1053	8.5	20.6	7.2	9.7	3938	12.4	24.1	11.6	13.1	3417	15.3	26.9	14.4	16.2	<0.001	all different
sf36-bp	1058	21.5	13.3	20.7	22.3	3974	24.4	14.2	23.9	24.8	3461	25.9	14.0	25.4	26.3	<0.001	all different
sf36-gh	1052	43.1	21.0	41.8	44.4	3944	42.6	20.5	42.0	43.2	3424	42.3	20.3	41.7	43.0	0.556	NA
sf36-vt	1058	21.4	17.8	20.3	22.5	3972	24.7	19.1	24.1	25.3	3452	26.3	18.6	25.7	26.9	<0.001	all different
sf36-sf	1056	42.4	24.2	40.9	43.9	3976	48.2	25.6	47.4	49.0	3458	51.9	25.2	51.0	52.7	<0.001	all different
sf36-re	1046	32.1	40.3	29.7	34.5	3906	44.0	43.2	42.7	45.4	3363	49.1	43.2	47.6	50.5	<0.001	all different
sf36-mh	1058	49.6	20.9	48.3	50.9	3970	56.0	21.0	55.3	56.7	3451	58.7	20.9	58.0	59.4	<0.001	all different

LB = Lower Bound; UB = Upper Bound; NA = not applicable; NRS-7days = Pain intensity as measured by a numeric rating scale for the previous seven days; HADS = Hospital Anxiety and Depression Scale; MPI = Multidimensional Pain Inventory; EQ-5D-index = The index of the European quality of life instrument; EQ-VAS = The European quality of life instrument thermometer-like scale; sf36 = The Short Form (36) Health Survey; PRI = Pain Region Index. For explanations of the subscale abbreviations, see Methods.

**Table 10 jcm-08-00905-t010:** Orthogonal Partial Least Square Regressions (OPLS) regressions of MIS post-IMMRP (left part) and at 12-month FU (right part) using the variables pre-IMMRP as regressors. Variables in bold type are significant regressors.

Post-IMMRP	VIP	p(corr)	12-Month FU	VIP	p(corr)
Variables Pre-IMMRP			Variables Pre-IMMRP		
**sf36-mh**	**1.80**	**−0.80**	**sf36-mh**	**1.63**	**−0.62**
**MPI-Distress**	**1.72**	**0.76**	**MPI-Distress**	**1.59**	**0.61**
**HADS-D**	**1.56**	**0.68**	**sf36-sf**	**1.44**	**−0.53**
**MPI-LifeCon**	**1.48**	**−0.65**	**sf36-re**	**1.39**	**−0.54**
**HADS-A**	**1.48**	**0.65**	**MPI-LifeCon**	**1.35**	**−0.49**
**sf36-sf**	**1.47**	**−0.63**	**HADS-D**	**1.34**	**0.46**
**sf36-re**	**1.43**	**−0.64**	**HADS-A**	**1.28**	**0.46**
**sf36-vt**	**1.27**	**−0.54**	**MPI-Pain-interfer**	**1.26**	**0.39**
**MPI-Pain-interfer**	**1.26**	**0.45**	**Persistent-Pain-duration**	**1.16**	**−0.46**
**EQ-5D-index**	**1.12**	**−0.41**	**Pain duration**	**1.15**	**−0.45**
**sf36-bp**	**1.07**	**−0.35**	**EQ-5D-index**	**1.14**	**−0.37**
**MPI-Pain-severity**	**1.05**	**0.28**	**sf36-bp**	**1.11**	**−0.35**
**EQ-VAS**	**1.04**	**−0.39**	**sf36-vt**	**1.11**	**−0.36**
sf36-gh	0.95	−0.30	MPI-Pain-severity	0.99	0.21
NRS-7days	0.88	0.20	EQ-VAS	0.97	−0.28
sf36-pf	0.87	−0.07	sf36-rp	0.96	−0.37
sf36-rp	0.80	−0.33	PRI	0.95	−0.26
PRI	0.72	−0.09	sf36-gh	0.87	−0.10
MPI-GAI	0.68	−0.20	NRS-7days	0.81	0.11
MPI-punish	0.59	0.25	Days no work	0.80	-0.31
Outside-Europe	0.45	−0.01	sf36-pf	0.76	−0.05
Days no work	0.44	−0.16	Age	0.75	−0.30
University	0.43	0.12	MPI-GAI	0.72	−0.22
Persistent Pain duration	0.42	−0.16	University	0.54	0.18
MPI-protect	0.41	−0.04	MPI-punish	0.43	0.12
Pain duration	0.38	−0.15	Outside-Europe	0.39	−0.03
MPI-Socsupp	0.32	−0.05	MPI-protect	0.31	0.03
MPI-distract	0.29	−0.01	MPI-distract	0.26	0.06
Age	0.23	−0.10	MPI-Socsupp	0.22	0.07
Gender	0.04	−0.01	Gender	0.10	−0.04
R^2^	0.08		R^2^	0.08	
Q^2^	0.08		Q^2^	0.07	
*n*	14 657		*n*	7 976	
CV-ANOVA *p*-value	<0.001		CV-ANOVA *p*-value	<0.001	

VIP (VIP > 1.0 is significant) and p (corr) are reported for each regressor. The sign of p (corr) indicates the direction of the correlation with the dependent variable (+ = positive correlation; − = negative correlation). The four bottom rows of each regression report R^2^, Q^2^, and *p*-value of the CV-ANOVA and number of patients included in the regression (*n*). NRS-7days = Pain intensity as measured by a numeric rating scale for the previous seven days; HADS = Hospital Anxiety and Depression Scale; MPI = Multidimensional Pain Inventory EQ-5D-index = The index of the European quality of life instrument; EQ-VAS = The European quality of life instrument thermometer-like scale; sf36 = The Short Form (36) Health Survey; PRI = Pain Region Index. For explanations of the subscale abbreviations see Methods.

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
