# Peer review of "Moderate and Stable Pain Reductions as a Result of Interdisciplinary Pain Rehabilitation—A Cohort Study from the Swedish Quality Registry for Pain Rehabilitation (SQRP)"

_jcm, 2019, doi:10.3390/jcm8060905_

Reviewer 1 Report

Thank you for opportunity for reviewing this interesting paper. The research adhere to reporting STROBE guidelines. This paper provides useful information on evaluate the prevalence of sociodemographic factors with the presence of chronic pain. If conducted with academic rigor, this article has the potential to be of value for physician and policymakers around the prevalence, cost and prevention of complications for people with chronic pain.

Furthermore, in my opinion the topic and premise of the study would sit well within the journal to which it was submitted. The authors should be commended for undertaking this study. Of particular note, have identified aspects of the written English as needing improvement. I would therefore recommend you get additional assistance on this aspect of the manuscript. We suggest you have a fluent, preferably native, English-language speaker thoroughly copyedit your manuscript for language usage, spelling, and grammar. If you do not know anyone who can do this, you may wish to consider employing a professional scientific editing service.

Also, there are a major concerns with the manuscript that require attention prior to publication. These will be discussed below relative to the sections of the manuscript.

TITLE

The title of this manuscript are a little long. Perhaps a more concise version for clarity, interes and ease of read.

ABSTRACT

It is hard to get the detail in an abstract when the word count is limited and this is often the hardest part of a paper to write. However, I do feel that it would be beneficial to explain what specifically you are looking at in relation to pathology (this also applies to the main body of the paper). Is it the development of chronic pain   literature.  This needs to be made clearer throughout the paper.

KEYWORDS:

Please use recognised MeSH terms as this will assist others when they are searching for information on your research topic. The following website will provide these (simply start typing in a keyword and see if it exists or find an alternative if it does not): https://www.ncbi.nlm.nih.gov/mesh

INTRODUCTION:

The introduction is weak. An introduction should announce your topic, provide context and a rationale for your work, while catching the reader´s interest and attention. The above has not been given in the introduction that I have read.

Thus, I suggest in this section should be improved, with more details about prevalence, impact related with chronic pain. Also, please describe functional limitations associated with chronic pain.

Finally, please describe the hypothesis and objectives in this section.

MATERIAL AND METHODS:

This section is poor, needs to present a better rationale for the study and the methodology employed. Also, neither appear information related with inclusion and exclusion criteria, dates, protocol.

Please, expand and clarification information related with the Swedish Quality Registry for Pain Rehabilitation, e Multidimensional Pain Inventory, Hospital Anxiety and Depression Scale, SF36 and European Quality of Life instrument  related with reliability and validity and the actual measurements.

Please describe in the text information related with this research for adhere to reporting STROBE guidelines.

Likewise more detail about information calculate sample size and data should be provided. Also, please need include the data and record code and all information related with the ethics committee and explain aspects ethics and legal requirement about this research.

RESULTS:

The results in basis of the used method are not informative. I dont believe this study adds a great deal of novel and new information.

DISCUSSION:

I am struggling to make sense of some of this, I am afraid it needs extensive revision. What are the clinical and non clinical implications of your study? How this will inform future larger studies?

Include this section the principal strengths and weaknesses in relation to other studies, discussing important differences in results; the meaning of the study: possible explanations and implications and unanswered questions and future research.

CONCLUSION:

These conclusions need to be softened, modified a in order to reflect only the study findings.

Author Response

Reviewer 1

Thank you for opportunity for reviewing this interesting paper. The research adhere to reporting STROBE guidelines. This paper provides useful information on evaluate the prevalence of sociodemographic factors with the presence of chronic pain. If conducted with academic rigor, this article has the potential to be of value for physician and policymakers around the prevalence, cost and prevention of complications for people with chronic pain.

Furthermore, in my opinion the topic and premise of the study would sit well within the journal to which it was submitted. The authors should be commended for undertaking this study.

Our answer: Thank you for considering our paper as interesting.
However, we do not think that the manuscript is adequately characterized by the reviewer in the following sentence: “This paper provides useful information on evaluate the prevalence of sociodemographic factors with the presence of chronic pain”.
Instead, we have investigated - as is evident from abstract, and at the end of the introduction - the following issues
in chronic pain patients in specialist care included in the Swedish Quality Registry for Pain Rehabilitation: effect sizes for 22 outcomes; analysed correlation patterns of outcome changes; defined a multivariate outcome measure; and investigate whether the clinical self-reported presentation before rehabilitation predicted the multivariate outcome measure.

Of particular note, have identified aspects of the written English as needing improvement. I would therefore recommend you get additional assistance on this aspect of the manuscript. We suggest you have a fluent, preferably native, English-language speaker thoroughly copyedit your manuscript for language usage, spelling, and grammar. If you do not know anyone who can do this, you may wish to consider employing a professional scientific editing service.

Our answer: The submitted manuscript was reviewed with respect to English by native speaking persons working at a professional and authorized translation agency in Sweden (Accent SprĂĄkservice AB; webpage: https://accent-sweden.com/). We have used them for more than 15 years. Accent SprĂĄkservice have contracts with several medical faculties in Sweden. Hence, they are specialized in English medical literature.

Also, there are a major concerns with the manuscript that require attention prior to publication. These will be discussed below relative to the sections of the manuscript.

TITLE

The title of this manuscript are a little long. Perhaps a more concise version for clarity, interes and ease of read.

Our answer: The title is shortened and very briefly summarizes important results of our study. The title is now: “Moderate and stable pain reductions as a result of Interdisciplinary Pain Rehabilitation – a cohort study from the Swedish Quality Registry for Pain Rehabilitation (SQRP)”

ABSTRACT

It is hard to get the detail in an abstract when the word count is limited and this is often the hardest part of a paper to write. However, I do feel that it would be beneficial to explain what specifically you are looking at in relation to pathology (this also applies to the main body of the paper). Is it the development of chronic pain   literature.  This needs to be made clearer throughout the paper.

Our answer: There must be some kind of misunderstanding here. Our study is not related to the pathology of chronic pain. It is a study concerning outcomes of interdisciplinary rehabilitation in patients with chronic pain conditions with special focus upon the complex interactions between outcomes.

KEYWORDS:

Please use recognised MeSH terms as this will assist others when they are searching for information on your research topic. The following website will provide these (simply start typing in a keyword and see if it exists or find an alternative if it does not): https://www.ncbi.nlm.nih.gov/mesh

Our answer: We now have revised the keywords and included the following MeSH terms: “Chronic Pain; Musculoskeletal Pain; Patient Care Team; Rehabilitation; Treatment Outcome;”

INTRODUCTION:

The introduction is weak. An introduction should announce your topic, provide context and a rationale for your work, while catching the reader´s interest and attention. The above has not been given in the introduction that I have read.

Thus, I suggest in this section should be improved, with more details about prevalence, impact related with chronic pain. Also, please describe functional limitations associated with chronic pain.

Finally, please describe the hypothesis and objectives in this section.

Our answer: In the revised version of the manuscript we have added our hypotheses: “We hypothesised that IMMRP in special care is associated with small to medium ES, that changes in outcomes generally are intercorrelated and that the baseline situation (pre- IMMRP) can predict the multivariate outcomes.”
The objectives/aims were clearly described at the end of the Introduction in the submitted version of the manuscript in four bullet points.

In the first paragraph of the Introduction we now have added two sentences (including two new references) concerning prevalence and consequences of chronic pain: “One-fifth of the European population has moderate to severe chronic pain conditions [4]. These conditions are associated with psychological distress, low health, sick leave, and high socioeconomic costs [5].

The reviewer points out that an introduction should announce your topic, provide context and a rationale for your work, while catching the reader´s interest and attention; we of course agree on a general level with this. However, the reviewer concludes that this has not been given in our introduction. We do not agree with this specific conclusion. Hence, we clearly announced our topics, provided context and a rationale for our work. Moreover, the second reviewer has no criticism of the introduction.
Our introduction consists of five paragraphs. In the first paragraph is briefly described the complexity of chronic pain and the need to adopt a bio-psycho-social framework (and as mentioned above also have added figures about the prevalence and the impacts of chronic pain conditions). In the second paragraph is described the general contents of interdisciplinary multimodal rehabilitation programs (IMMRP). In the third paragraph is briefly reported the evidence for IMMRP based upon RCTs and SRs. Moreover, we point out the need to replicate the evidence from RCTs and SRs within the consecutive non-selected flow of patients in practice settings. Moreover, we point out that it is unclear if pain intensity is affected by IMMRP according to SRs but that there are studies in fact presenting results indicating effects upon pain intensity. We also mention that this an important aspect of outcome for many patients. In the fourth paragraph we concluded that the outcomes of IMMRP often are several and that these outcomes most likely are intercorrelated but despite this several SRs evaluate the outcomes independent from each other. We conclude that there is a need to develop clinically applicable ways to evaluate multiple outcomes of MMPRs both for individual patients and in research studies. In the fifth paragraph we present our four aims and the fact that the Swedish Quality Registry for Pain Rehabilitation (SQRP) SQRP offers an opportunity to investigate these aspects within the consecutive non-selected flow of patients in practice settings within specialist care.

MATERIAL AND METHODS:

This section is poor, needs to present a better rationale for the study and the methodology employed. Also, neither appear information related with inclusion and exclusion criteria, dates, protocol.

Our answer: The rationales are (and should be) given in the Introduction. In the first paragraph is mentioned that SQRP is based on questionnaires. Due to the comment concerning inclusion criteria we have added the following: “Strict inclusion and exclusion criteria for inclusion in the registry is not available since this is a registry study of patients with complex chronic pain conditions referred from mainly the primary care to specialist care in Sweden; a minority of patients are referred from other specialist clinics e.g. orthopedic and rheumatology clinics. The following general inclusion criteria for IMMRP were used: (i) disabling chronic pain (on sick leave or experiencing major interference in daily life due to chronic pain); (ii) age 18 years and above; (iii) no further medical investigations needed; (iv) written consent to participate and attend IMMRP. General exclusion criteria for IMMRP included severe psychiatric morbidity, abuse of alcohol and/or drugs, diseases that did not allow physical exercise, and specific pain conditions with other treatment options available (i.e., red flags).”

Please, expand and clarification information related with the Swedish Quality Registry for Pain Rehabilitation, e Multidimensional Pain Inventory, Hospital Anxiety and Depression Scale, SF36 and European Quality of Life instrument  related with reliability and validity and the actual measurements.

Our answer: As to our answer above we have added information about how patients are referred to the specialist care and included in SQRP. Moreover, due to this comment of the reviewer we also have added the following information “The proportions of patients within primary health care with chronic pain conditions are not exactly known but 10-20% are estimates [32, 33]. Furthermore, the proportion of chronic pain patients within primary health care that are referred to specialist clinics is not known.”

It is somewhat unclear written (words missing) but we interpret it as information about the psychometric properties are needed. In order to not expand the number of references too much we have referred to previous work reporting such aspects: “For reports of the psychometric aspects of the self-reported measures the reader is referred to other studies summarizing these [7, 34-36].”

Please describe in the text information related with this research for adhere to reporting STROBE guidelines.

Our answer: We now have added a protocol reporting STROBE for our study.

Likewise more detail about information calculate sample size and data should be provided. Also, please need include the data and record code and all information related with the ethics committee and explain aspects ethics and legal requirement about this research.

Our answer: It must be a misunderstanding with respect to sample size calculations. We have used data from a national registry and all patients fulfilling our requirements were included in our study. Hence, in such a situation it is not relevant to calculate sample size. In fact, we have several thousand patients and a more important issue is to determine if the significant changes also are clinically relevant. In order to do that we have presented the effect sizes (ES) for the 22 outcomes.

In the section of Declarations of the submitted manuscript we reported: “The study was conducted in accordance with the Helsinki Declaration and Good Clinical Practice and approved by the Ethical Review Board in Linköping (Dnr: 2015/108-31). All participants received written information about the study and gave their written consent.” We now instead have moved these sentences to the section of subjects in the revised manuscript.

RESULTS:

The results in basis of the used method are not informative. I dont believe this study adds a great deal of novel and new information.

 Our answer: This is an unspecific comment. We note that reviewer 2 not had any such comments. Moreover, in our opinion it is confusing in relation to the general comment given in the beginning of the reviewer i.e. “Thank you for opportunity for reviewing this interesting paper.” We have not made any changes due to this comment.

DISCUSSION:

I am struggling to make sense of some of this, I am afraid it needs extensive revision. What are the clinical and non clinical implications of your study? How this will inform future larger studies?

Include this section the principal strengths and weaknesses in relation to other studies, discussing important differences in results; the meaning of the study: possible explanations and implications and unanswered questions and future research

Our answer: We have due to this comment added a paragraph concerning clinical implications:

“Important clinical implications

Outcomes of IMMRP in real-life practice settings agree with the conclusions from SRs. Partly in contrast to SRs this registry study of patients managed within specialist care found that pain intensity was positively affected because of IMMRP. It was also obvious that not all patients benefit from IMMRP. Hence, there is a need to develop better matching between clinical presentation and participation in MMRP in real-life practice settings. Moreover, the intercorrelations of most changes in outcomes also opens up for the possibility of reducing the number of outcome variables and herby reduce the burden upon patients included in SQRP. “
We doubt there is a need for larger studies (more than 14 000 patients were included in the present study).

We have discussed our results in relation to other studies including SRs in every paragraph of the Discussion (except in the paragraphs of Important clinical implications, Strengths and Limitations and Conclusions) in the submitted manuscript. Moreover, we note that reviewer 2 not at all had such general comments concerning the Discussion. Instead we have perceived that he/she was satisfied with the Discussion.

CONCLUSION:

These conclusions need to be softened, modified a in order to reflect only the study findings.

Our answer: Due to this comment we have omitted the following sentence: “These results are in line with RCTs and SRs and advocate for IMMRP as a meaningful intervention for complex chronic pain patients.”

Reviewer 2 Report

Thank you for this interesting and important study.

There are many abbreviations in the text, so it can be good with a list of abbreviations.

There are four main aims. Does it exists an overall aim that describe these four specific aims?

The data were collected during nine years, i.e., between 2008 and 2016. Did the IMMRP change during these years? I lack a discussion about the strengths and weaknesses with a long period for data collection.

“14 666 chronic pain patients registered in SQRP that fulfilled the inclusion criteria: 252 chronic pain; >18 years of age; and completed the SQRP questionnaire before and on at least one of the two time points after the IMMRP”. Please, describe the response rate at time point 1 and time point 2.

Author Response

Reviewer 2

There are many abbreviations in the text, so it can be good with a list of abbreviations.

 Our answer: We now have added a list of abbreviations.

There are four main aims. Does it exists an overall aim that describe these four specific aims?

Our answer:  We now have inserted a general aim also: “Hence, this PBE study has the general aim of investigating the effects of IMMRP in specialist care in Sweden considering the multivariate complexity of outcomes. We hypothesised that IMMRP in special care is associated with small to medium ES, that changes in outcomes generally are intercorrelated and that the baseline situation (pre- IMMRP) can predict the multivariate outcomes. More specifically we defined the following four aims:”

Due to comments of the first reviewer we have added an overall hypothesis.

The data were collected during nine years, i.e., between 2008 and 2016. Did the IMMRP change during these years? I lack a discussion about the strengths and weaknesses with a long period for data collection.

Our answer: Thank you for this constructive comment. We now in the section of Strengths and limitations have added: “Data for the time period 2008-2016 from SQRP was used in the present study and changes in the content of IMMRP may have occurred. Unfortunately, no data concerning such changes are available.”

“14 666 chronic pain patients registered in SQRP that fulfilled the inclusion criteria: 252 chronic pain; >18 years of age; and completed the SQRP questionnaire before and on at least one of the two time points after the IMMRP”. Please, describe the response rate at time point 1 and time point 2.

Our answer: The total number of patients referred to specialist clinics in Sweden is not known, but the steering committee for SQRP has estimated the response rate of >90% for complex chronic pain conditions. A subgroup of these patients registered in SQRP will participate in IMMRP. In a previous article we have reported that approximately 54% participate in IMMRP. 60% of those answering the PROM questionnaires pre- IMMRP and post-IMMRP also answered the questionnaires at 12-m FU. In the first paragraph of the Results we now have added the following: “60% of the patients answering the questionnaires pre- IMMRP and post-IMMRP also answered the questionnaires at 12-m FU.”

Due to comments of the first reviewer we also have added more information about SQRP in the section of methods.

Round  2

Reviewer 1 Report

The authors have satisfactorily responded to all of my comments.